# Associations of Maternal Serum Iodine Concentration with Obstetric Complications and Birth Outcomes—Longitudinal Analysis Based on the Huizhou Mother–Infant Cohort, South China

**DOI:** 10.3390/nu15132868

**Published:** 2023-06-25

**Authors:** Zhao-Min Liu, Yi Wu, Huan-Huan Long, Chao-Gang Chen, Cheng Wang, Yan-Bin Ye, Zhen-Yu Shen, Ming-Tong Ye, Su-Juan Zhang, Min-Min Li, Wen-Jing Pan

**Affiliations:** 1Guangdong Provincial Key Laboratory of Food, Nutrition and Health, Department of Nutrition, School of Public Health, Sun Yat-sen University, North Campus, Guangzhou 510080, China; wuyi36@mail2.sysu.edu.cn (Y.W.); longhh5@mail2.sysu.edu.cn (H.-H.L.); zhangsj25@mail2.sysu.edu.cn (S.-J.Z.); limm9@mail2.sysu.edu.cn (M.-M.L.); 2Department of Clinical Nutrition, Second Affiliated Hospital, Sun Yat-sen University, Guangzhou 510080, China; chenchg@mail.sysu.edu.cn (C.-G.C.); wangcheng@mail.sysu.edu.cn (C.W.); 3Department of Clinical Nutrition, The First Affiliated Hospital, Jinan University, Guangzhou 510630, China; yeyanbin2023@jnu.edu.cn; 4Department of Paediatrics, The First Affiliated Hospital, Sun Yat-sen University, Guangzhou 510080, China; shenzhy@mail.sysu.edu.cn; 5Huizhou First Maternal and Child Health Care Hospital, Huizhou 516000, China; timyale@126.com

**Keywords:** serum iodine, gestational diabetes mellitus, maternal thyroid function, gestational hypertension, birth weight, small and large for gestation age

## Abstract

This study aimed to explore the temporal associations between maternal serum iodine concentration (SIC) and common pregnancy outcomes in Chinese women. Eligible singleton pregnant women aged 20–34 years were selected, and their fasting blood samples were collected during early (T1, n = 1101) and mid-pregnancy (T2, n = 403) for SIC testing by inductively coupled plasma mass spectrometry. Multivariable linear regression indicated that log_10_SIC at T1 (β = −0.082), T2 (β = −0.198), and their % change (β = −0.131) were inversely associated with gestational weight gain (GWG, all *p* < 0.05). Maternal log_10_SIC at both T1 (β = 0.077) and T2 (β = 0.105) were positively associated with the Apgar score at 1 min (both *p* < 0.05). Women in the third quartile (Q3) of SIC at T1 had a lower risk of small for gestational age (SGA, OR = 0.405, 95% CI: 0.198–0.829) compared with those in Q4. Restricted cubic spline regression suggested a U-shaped association between SIC and SGA risk, and SIC above 94 μg/L at T1 was the starting point for an increased risk of SGA. The risk of premature rupture of membrane (PROM) increased by 96% (OR = 1.960, 95% CI: 1.010–3.804) in Q4 compared to that in Q1. Our longitudinal data from an iodine-replete region of China indicated that high maternal SIC could restrict GWG and improve Apgar scores at delivery, but might increase the risk of SGA and PROM.

## 1. Introduction

Iodine is one essential ingredient of thyroid hormones. Iodine requirements during gestation increase sharply due to elevated hormone production, fetal needs, and renal excretion [1]. Iodine excess has become prevalent in recent years and attracts a lot of concern because of extensive iodine prescription during gestation and universal salt iodization even in iodine-replete regions [2]. Pregnant women and neonates are susceptible to both iodine deficiency and excess, both of which were associated with poor pregnancy outcomes [3]. Early pregnancy is a particularly vulnerable period for iodine nutrition since maternal iodine is the only source of fetal thyroid hormone synthesis [4]. However, the potential influence of excessive iodine exposure has not been well studied, especially during gestation.

WHO recommends urinary iodine concentration (UIC) for evaluation of populational iodine status, but not applicable to individual iodine assessment [5]. Multiple days urinary iodine excretion (UIE) is supposed to be an ideal indicator for assessment of personal iodine status, but unpracticable in epidemiological investigations due to the cumbersome urine collection requirements. Observational studies employing spot UIC for the assessment of nutritional iodine status reported conflicting findings on its relationship with obstetric complications and pregnancy outcomes [6,7,8] even if additional adjustment for urinary creatinine (UCr) was performed [9]. This could be due to UIC varying greatly with actual dietary iodine intake, season, and circadian rhythmicity [10], which could lead to the misclassification of iodine status and compromise the difference among different UIC groups. Accurate assessment of iodine nutrition is therefore essential, particularly during pregnancy [11].

Serum iodine concentration (SIC) has been considered the most accurate and reliable biomarker for assessing bioavailable iodine in the thyroid gland due to its better stability and fewer between-individual variations than UIC [12]. However, longitudinal studies employing SIC as a biomarker and exploring its relationship with pregnancy outcomes have been much more limited. The optimal maternal SIC has not been clarified for birth outcomes. We thus conducted a longitudinal analysis on the basis of a prospective birth cohort in Huizhou, a coastal city in an iodine-sufficient area of South China. We used trimester-specific SIC as a clinical biomarker to assess individual iodine status for pregnant women and explored its relationship with common obstetric complications and birth outcomes.

## 2. Methodology

### 2.1. Participant Recruitment

This was a longitudinal study embedded in the Huizhou Mother–Infant Cohort which was a joint project by the School of Public Health of Sun Yat-sen University and Huizhou First Maternal and Children’s Hospital (HFMCH). The study protocol was approved by the Ethics Committee of HFMCH (registration no. 2018002). All the participants signed written consent forms before enrollment. 

Singleton pregnant women aged 20–34 years regularly residing in Huizhou City were selected from the Huizhou Mother–Infant Cohort during their first antenatal visit from September 2020 to June 2021 with the original purpose of establishing trimester-specific reference intervals for SIC. Women were excluded if they were currently taking iodine-containing drugs, under medication for thyroid dysfunction, or subject to high vocational iodine exposure; had a medical history of thyroid dysfunction or other severe endocrinal disorder; currently had an autoimmune disease, or severe kidney or liver dysfunction; or their venous samples from early pregnancy were unavailable. Participants were asked to complete a questionnaire and donate spot urine samples for iodine testing in their first prenatal visit. The study flow chart was shown in Figure 1. In order to reduce the burden of specimen collection for the participants, we used leftover serum samples for iodine testing which were collected in a timely manner after routine clinical biochemical testing. The original purpose of the study was to establish the trimester-specific reference intervals of SIC for Chinese pregnant women. The adverse pregnancy outcomes were thus studied among generally healthy women with live births.

### 2.2. Data Collection and Biochemical Testing

Individual information, including socio-demographics, health and obstetric history, family history of common chronic diseases, medication treatment, and lifestyle factors (i.e., smoking, alcohol drinking, physical activities, usage of iodine-fortified salt, and dietary consumption of iodine-rich foods) was collected via face-to-face interviews using a pretested questionnaire. The anthropometrics of maternal weight and height were measured during early pregnancy and before delivery by a standard method with outer garments and shoes being removed. Pre-pregnant body weight (the most recent measure before pregnancy) was self-reported. Body mass index (BMI) was calculated by dividing body weight (kg) by squared height (m^2^). Birth weight and length were measured by trained midwives in the delivery room with standard methods. The ponderal index (PI) was estimated by dividing birth weight (kg) by cubic length (m^3^).

Fasting (8–12 h overnight) venous samples were collected during early (10–12 weeks, n = 1101) and mid-pregnancy (24–28 weeks, n = 403) for determining SIC. The sample size for SIC testing was estimated in consideration of the original study purpose and available funding budget. In order to avoid iodine contamination, alcohol instead of iodophor disinfection was used and all the investigators wore masks to prevent extra iodine contamination from saliva. Blood samples were centrifuged at 3000 rpm/min for 10 min at 4 °C, then aliquoted and stored in a −80 °C freezer until analysis. Spot urine samples were voluntarily donated by 390 women during early pregnancy. The mid-stream urine was allocated into several 2 mL vials with no additives after complete blending and stored in a −26 °C freezer for measuring UIC and urinary creatinine (UCr) levels.

Maternal SIC and UIC were tested by inductively coupled plasma mass spectrometry (ICP-MS, Agilent 7700x, Agilent Technologies, Inc., Santa Clara, CA, USA) [13] and handled according to Chinese National Hygiene Standards WS/T 783-2021 and WS/T 107.2-2016, respectively. The iodine standards (GSB 04-2834-2011, ClinChek^®^ Serum Control, RECIPE, Munich, Germany) were tested intermittently for quality control (every 20th specimen). Serum samples were diluted by a mixture solution of ascorbic acid–ammonium chloride–ethanol amine–ethanol before analysis and analyzed with the use of rhenium for mass bias correction. Thyroid hormones including free thyroxine (FT4) and thyroid stimulating hormone (TSH) were tested in early pregnancy by electro-chemiluminescence assay. Plasma glucose and urinary creatinine levels were analyzed by the color-enzymic and hexokinase methods, respectively, on an autoanalyzer (Cobas c702, Roche Diagnostics Ltd., Indianapolis, IN, USA). All the intra- and inter-assay coefficients of variation (CVs) of the biochemicals were less than 5%.

### 2.3. Health Outcomes and Diagnosis Criteria

We examined the following obstetric complications and birth outcomes: gestational diabetes mellitus (GDM), gestational weight gain (GWG), gestational hypertension (GH), premature rupture of membrane (PROM), preterm birth, postpartum hemorrhage, abnormal amniotic fluid (<300 mL or >2000 mL), Apgar scores after delivery at 1 min, 5 min, and 10 min, and birth weight and length (low birth weight (LBW) and small and large for gestational age (SGA and LGA)). All the clinical information was retrieved from the Hospital Information System. 

A standard 2 h 75 g oral glucose tolerance test (OGTT) was performed at 24–28 GWs for GDM diagnosis. Participants were asked not to drink tea or coffee or do strenuous exercises before or during the test. GDM was diagnosed in accordance with the recommendation of the International Association of Diabetes and Pregnancy Study Groups (IADPSG) if at least one of the glucose levels was above the cut-off: 0 h ≥5.1 mmol/L, 1 h ≥10.0 mmol/L, or 2 h ≥8.5 mmol/L [14]. The incidence of GH was based on medical records and defined as systolic BP ≥ 140 mmHg and/or diastolic BP ≥ 90 mmHg on two or more occasions beyond 20 GWs. GWG was calculated as the difference between the self-reported pre-pregnancy weight and the last clinically measured weight before delivery, and categorized as inadequate, appropriate, or excessive according to the 2021 Chinese Nutrition Society guidelines for GWG [15]. Preterm delivery was defined as a live birth before 36 completed weeks of gestation. An adverse birth outcome of SGA or LGA was defined as a birth weight less than 10% or more than 90% of the average birth weight, respectively, which were customized for gender and gestational age in accordance with established criteria in the Chinese population (2020) [16]. Post-partum hemorrhage was defined by an estimated blood loss of over 500 mL around the time of delivery.

### 2.4. Statistical Analysis

SPSS 25.0, Amos 25, Medcalc, and R 4.1.3 were applied for statistical analyses with significance being defined at *p* < 0.05. Data distribution and variance heterogeneity were tested by the Kolmogorov–Smirnov method. SICs at the first (T1) and second (T2) trimesters were log_10_-transformed due to skewed distribution and presented as medians (interquartile ranges, P_25_~P_75_). Multivariable logistic regression was applied to explore the associations of maternal SIC at T1 (quartiles) with the risk of GDM, GH, PROM, SGA, and LGA with either the lowest quartile (for GDM, GH, and PROM) or the highest quartile (for SGA and LGA) as the reference groups. Potential confounders were determined on the basis of the results of univariate regression, literature review, and potential biological mechanisms. For the logistic regression on PROM, mothers with macrosomia or polyhydramnios (>2000 mL) were excluded, and 788 women remained for analysis. Linear regression was applied to the associations of SIC at T2 and SIC % change from T1 to T2 with various pregnancy outcomes due to the possibly inadequate power for logistic models (n = 402 for SIC at T2, and insufficient events of adverse outcomes in a generally healthy population). Given the non-linearity of SIC at T1 with the risk of SGA, multivariable-adjusted regression of restricted cubic spline (RCS) was explored to flexibly the model and visualize the relationship of maternal SIC at T1 with the risk of SGA. Knots were located at the 5th, 50th, and 95th percentiles of SIC to evaluate the individual spline term contributions to the model fit, and ANOVA was used for non-linearity testing. Multivariable linear regression was used to examine the relationship of logarithm-transformed maternal SICs at T1 and T2 as well as their change % from T1 to T2 with pregnancy outcomes evaluated by continuous variables, including the sum of Z-scores by OGTT, Apgar scores, GWG, birth weight and length, as well as the ponderal index (PI). The sum of Z-scores by OGTT was a composite measure for gestational glycemic control and was calculated by adding all the individual Z-scores of the fasting, 1-, and 2-h postprandial glucose levels. For the GWG data, 8 irrational or extreme values were removed for analysis. Sensitivity analyses were conducted to verify the results’ consistency by the exclusion of women of preterm delivery, or additional adjustment of TSH level in the linear regression models between SIC and GWG. Additional analyses on the associations of UI/UCr with various pregnancy outcomes were performed. The interactions of SIC with thyroid hormones (FT4 or TSH) were also explored to verify the possible joint role of SIC at T1 with thyroid hormones in pregnancy outcomes (SGA and PROM).

## 3. Results

### 3.1. Baseline Characteristics

A total of 1101 pregnant women were tested for SIC during early pregnancy (10–12 GWs), of which 403 were randomly selected for mid-trimester SIC testing, 968 had complete data of fasting and post-load glucose levels at 24–28 GWs, and 809 neonatal birth outcomes were available from the Hospital Information System. The demographics and clinical characteristics of the pregnant women are shown in Table 1. Their average age, pre-pregnancy body mass index (BMI), and GWG were 28.1 ± 3.3 years, 20.8 ± 3.1 kg/m^2^, and 13.5 ± 4.6 kg, respectively. More than half of them (54.4%) were primiparous. The median (IQR, P_25_ to P_75_) SICs at 10–12 GWs and 24–28 GWs were 87.6 (77.4, 97.7) and 77.4 (71.6, 87.6) μg/L, respectively. According to WHO criteria for UIC classification, the iodine status of our participants was classified as adequate or above adequate with a median UIC of 190.6 μg/L (IQR: 131.2–260.0 μg/L). The incidence rates of GDM, GH, PROM, SGA, and LGA were 16.8% (169/1004), 2.0% (16/809), 11.7% (95/809), 12.4% (100/809), and 2.5% (20/809), respectively. Appendix A compares the baseline characteristics of women who were included for analysis during early pregnancy (n = 1101) with those being followed up till delivery (n = 809). The results suggested they were mostly comparable in terms of age, prepregnant BMI, education attainment, parity, smoking, and alcohol drinking.

### 3.2. Results of Multivariable Logistic and Linear Regression

Appendix A compared the baseline characteristics of mothers with delivery of infants of AGA with SGA. After controlling for potential confounders, multivariable logistic regression (Table 2) showed a null association between maternal SIC during early pregnancy and the risk of GDM, GH, and LGA. Compared with the lowest quartile of SIC, women in the highest group had a significantly increased risk of PROM (OR = 1.960, 95% CI: 1.010–3.804; *P*_trend_ = 0.103). Women in the third quartile (Q3) of SIC had the lowest risk of SGA (OR = 0.405, 95% CI: 0.198–0.829) in comparison with the highest quartile group (Q4). The plot of RCS regression (Figure 2) indicated a slightly ‘U-shaped’ association between maternal SIC in early pregnancy and the risk of SGA (*p* for non-linearity = 0.036). The risk of SGA decreased with the increase in maternal SIC (T1) until reaching 94 μg/L, and then gradually increased thereafter.

The results of multivariable linear regression are shown in Table 3. Logarithmic transformed SIC at T1 (*β* = −0.082, *p* = 0.025) and T2 (*β* = −0.198, *p* < 0.001), and the change % from T1 to T2 (*β* = −0.131, *p* = 0.019) were all significantly and inversely associated with GWG. The log_10_SICs at both T1 (*β* = 0.077, *p* = 0.026) and T2 (*β* = 0.105, *p* = 0.047) were positively associated with the Apgar score at 1 min. Although a null association was observed between maternal SIC at either T1 or T2 and birth weight and length, the increase in SIC change % from T1 to T2 was associated with decreased birth weight *(β* = −0.085, *p* = 0.054) and length (*β* = −0.092, *p* = 0.038) with marginal significance. Further adjustment of GWG attenuated these associations into null (*p* values from 0.054 to 0.157 for birth weight) or marginal significance (*p* values from 0.038 to 0.084 for birth length). No significant association was observed between maternal SIC at either T1 or T2 and other obstetric complications and outcomes, including the sum of Z-scores by OGTT, ponderal index, delivery weeks, the volume of postpartum hemorrhage, and Apgar scores at 5 or 10 min (all *p* > 0.05, data not shown). All additional analyses of the associations of UI/UCr with various pregnancy outcomes yielded non-significant findings (all *p* > 0.05, data not shown).

### 3.3. Sensitivity Analyses

No significant interactions were observed between the quartiles of SIC at T1 and high and low levels of FT4 or TSH, and the risk of PROM (*p* for the interaction of 0.134 and 0.330, respectively) or SGA (*p* for the interaction of 0.064 and 0.910, respectively). A sensitivity analysis among women with full-term deliveries (n = 772) showed similar associations as those of the participants as a whole (Appendix A). Further adjustment of maternal TSH attenuated the association between SIC and GWG (data not shown).

## 4. Discussion

### 4.1. Summary of Findings and Implications

Our longitudinal data from an iodine-replete region of China demonstrated that high maternal SIC might restrict gestational weight gain and improve Apgar scores at delivery but might increase the risk of SGA and PROM. The smaller reduction in SIC from T1 to T2 was associated with lowered GWG and birth size, suggesting that iodine excess during pregnancy might restrict maternal weight increase and fetal growth. The study utilized the trimester-specific SIC for the assessment of nutritional and bioactive iodine status and explored their relationship with multiple pregnancy outcomes in Chinese women. To our knowledge, this study was the first to observe a U-shaped association between maternal SI levels in the first trimester even in the normal range, and subsequent SGA risk, implying that both inadequate and excessive iodine levels might increase the risk of SGA. We are the first to define the safe upper limit of SIC (94 μg/L) during early pregnancy for the prevention of SGA. Our study in pregnant women provides reassuring new evidence that, relative to UIC, SIC is a sensitive and valid biomarker reflecting individual iodine status, which could reduce misclassification and improve statistical power in observational studies. The findings are useful for healthcare providers, facilitating reasonable prescription of iodine supplements during pregnancy to avoid excessive iodine exposure. Our study emphasized the crucial importance of SIC surveillance during gestation, esp. in mid-trimester among a population known to be susceptible to both iodine deficiency and excess. Meaningful findings were even observed among a generally healthy and euthyroid population. Future studies with the inclusion of mothers with various thyroid conditions and adverse pregnancy outcomes are necessary to substantiate the current findings.

### 4.2. SIC and Pregnancy Outcomes

Pregnant women living in iodine-sufficient regions commonly present with both iodine deficiency and excess [17], while the potential hazards of excessive iodine on pregnancy outcomes have not been adequately studied. Prior observational studies mostly focused on UICs to examine their relationship with pregnancy outcomes, and reported controversial findings. Two meta-analyses both reported non-significant associations of gestational UIC with various pregnancy outcomes (i.e., preterm birth, low birth weight, GH, Apgar score, etc.) [6] or anthropometric measures at birth [18]. The non-significant findings might be due to the intrinsically high variability in UIC which caused inadequate statistical power and compromised the comparisons between groups for UIC analysis. Only one nested case-control study conducted in Finland, a mildly iodine-insufficient country, investigated the relationship of maternal SIC in early gestation with pregnancy outcomes [19] and reported null associations of SIC with the risk of SGA, GDM, and GH, except for an increased risk of preterm birth. Although thyroid hormones play an important role in glucose metabolism and insulin secretion, we did not observe a significant association between maternal SIC and GDM risk or glycemia. This might be due to the fact that the participants in our analysis were mostly euthyroid, while mothers with thyroid dysfunction appeared to be more likely to develop GDM [20]. Our analysis observed positive associations of SIC during both early and mid-pregnancy with Apgar scores at 1 min, which was in line with previous reports that iodine supplementation or sufficient iodine status during pregnancy were associated with increased neonatal Apgar scores [21].

Our analysis indicated a null association between maternal SIC during early pregnancy and the risk of GH, although a marginally decreased trend was observed for the risk of GH by quartiles of SIC. To our knowledge, this is the first study reporting the association between SIC and GH in an iodine-sufficient area. Previous studies that verified the association between serum, urinary, or cord blood iodine levels and preeclampsia or hypertensive pregnancy disorders have drawn controversial conclusions [22,23,24,25,26]. High iodine level was reported to be associated with reduced oxidative stress and lowered risk of GH [22], as iodine, an antioxidant, might contribute to redox balance during pregnancy. The non-significant findings regarding the association between serum iodine and GH in our study could be due to the insufficient quantity of GH cases and the generally euthyroid women among our participants who might be insusceptible to vascular dysfunction. The iodine measuring time during gestation might also play a role. Studies that measured maternal iodine in the late trimester could not observe significant findings as clinical hypertension has already begun [27]. In addition, the major source of iodine was often iodized salt, and increased salt intake may increase the risk of hypertension and offset the favorable impact of iodine on GH. Future prospective studies with the inclusion of adequate GH incidence are necessary to confirm the current findings. 

Our analysis found an increased risk of PROM in the highest quartile of SIC during early pregnancy, and the association was more evident after the exclusion of women with preterm delivery. Although we adjusted the range of possible confounders in the multivariable logistic regression model, residual confounding was unavoidable as we did not investigate infection or trauma before or at delivery, the diseased condition of the membrane itself, and the polymorphisms of genes involved in hemostasis and angiogenesis contributing to PROM [28]. Although there has been no report on SIC and PROM, studies on maternal thyroid dysfunction and the risk of PROM have reported inconsistent findings [29,30]. The findings remain to be validated in future prospective studies.

### 4.3. SIC and GWG

To our knowledge, this study was the first to explore the association between maternal SIC and gestational weight gain in an iodine-sufficient region of China. Our findings indicated that higher maternal SIC in both early pregnancy and the mid-trimester, and even the relatively smaller decrease in SIC during gestation, could restrict maternal weight gain, with the highest impact being observed for mid-trimester SIC. Iodine is an essential nutrient for regulating growth, development, and metabolism via the biosynthesis of thyroid hormones, which could directly affect energy metabolism and body weight changes. A retrospective study involving 503 pregnant women in Spain found a negative association between GWG and the Mediterranean dietary pattern, a diet rich in iodine from seafood and kelp [31]. Our analysis indicated that maternal SIC in the second trimester, although significantly reduced, had more direct and greater impact on reducing weight gain than SIC in the first trimester. Further pathway analysis indicated that TSH may mediate the association between SI and maternal body weight change. Weight gain, as a proxy of a proper diet, was reported to be protective against iodine deficiency [32]. A study of 462 Belgian mothers also reported that higher GWG was linked with lower placental iodine storage [33]. The exact physiological mechanisms linking iodine status with maternal weight gain need further clarification.

### 4.4. SIC and Birth Outcomes

Fetal growth was affected by a range of factors, not only maternal characteristics (i.e., age, adiposity, nutritional status, smoking habits, and placental and in utero environment) but also genetic heritability. Previous studies mostly focused on the influence of iodine deficiency on birth weight and often applied UIC as a biomarker for the assessment of maternal iodine status. Studies tested maternal SIC, and the adverse impacts of excessive iodine on birth size were limited. In line with a recent meta-analysis [34] and several previous findings with UIC as an iodine marker [8,26,35], our analysis also revealed a potentially U-shaped association between SIC and SGA, and RCS regression further identified SIC above 94 μg/μL as being associated with increased risk of SGA. Additional adjustment of GWG attenuated the associations into marginal or null significance, implying that the decreased maternal weight gain partly mediated the association between SIC and SGA. Overtreatment of GDM might restrict maternal weight gain and fetal development; however, additional adjustment of GDM status in the regression model made little difference to the overall findings. When iodine levels are excessive, it is possible that high maternal SICs may induce an increased risk of autoimmunity [36] and a decrease in fetal thyroid hormones [37], which affect fetal skeletal development and placentation, resulting in low birth weight or SGA. Studies have shown that both insufficient or excessive iodine intake during gestation may compromise the adaptive mechanisms in maternal thyroid function and lead to adverse pregnancy outcomes [38,39]. Maternal iodine might also impact birth outcomes through non-thyroidal mechanisms, including the regulation of oxidative stress, enzyme function, signal transduction, and transcriptional activity, especially during early pregnancy [40]. Further studies are needed to clarify the potential mechanisms.

### 4.5. Strengths and Limitations

The major strengths of the present study are the prospective study design, the relatively large sample size for trimester-specific SIC testing, and the comprehensive data collection for covariates, which together minimized the reversal and residual biases. SIC was detected by a validated method of ICP-MS [41] with good precision, high accuracy, and good recovery. In addition, the participants in our study were mostly healthy, euthyroid, and iodine-sufficient, which could exclude the influence of other nutritional deficiencies that might confound the association between SIC and birth outcomes. Because of the importance of mid-pregnancy SIC for GWG, the surveillance of SIC across gestation is necessary for the prevention of potential adverse pregnancy outcomes.

Several limitations merit discussion. First, most of our participants were healthy and euthyroid. Additional studies with the inclusion of mothers with thyroid dysfunction and adverse birth outcomes are warranted to substantiate the current findings and extend the study’s generalizability. Second, we did not investigate dietary intake and the supplemental usage of iodine, which may not be completely captured in SIC due to their volatility. Dietary seaweed and kelp intakes as well as the usage of iodized salt were reported by less than half of the participants, which limited our estimation of the exogenous iodine contributions to the SIC. In future studies, both SIC and iodine supplement intakes should be examined longitudinally throughout the gestational period to confirm these findings. Finally, because of the observational nature of the study, we cannot rule out the possibility of confounding from unmeasured factors such as dietary patterns, chronic thyroid conditions, or iodine supplementation initiated after blood collection.

## 5. Conclusions

Our longitudinal data from an iodine-replete region of China demonstrated that high maternal SIC could restrict gestational weight gain and improve Apgar scores at delivery, but might increase the odds of SGA and PROM. Future longitudinal studies with the inclusion of mothers with various thyroid dysfunctions and obstetric complications, and follow-up for infant and childhood neurodevelopment, are warranted.

## Figures and Tables

**Figure 1 nutrients-15-02868-f001:**
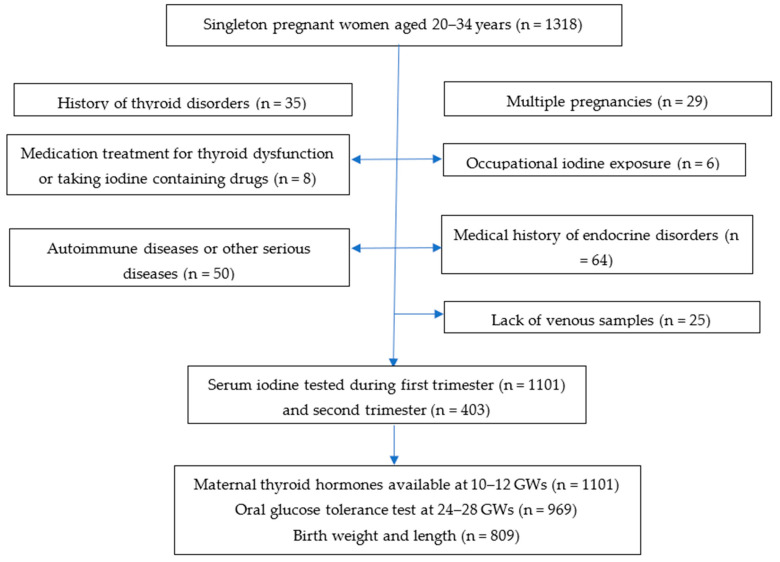
Study flow chart. Abbreviations: GW, gestational weeks.

**Figure 2 nutrients-15-02868-f002:**
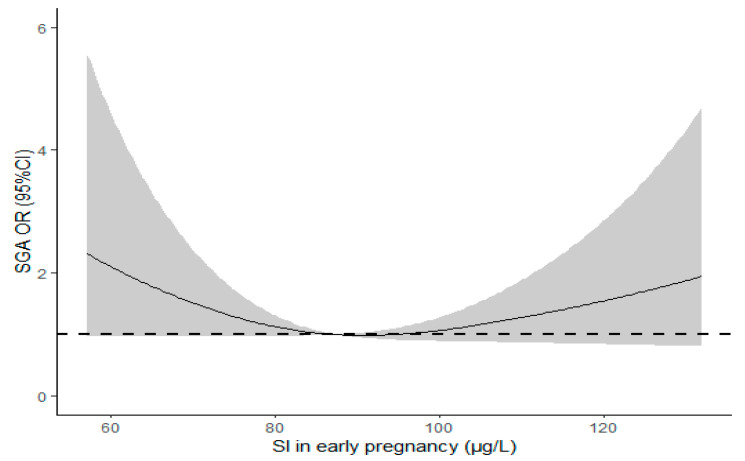
Restricted spline cubic (RCS) regression on the association between serum iodine concentration (SIC) and the risk of small for gestational age (SGA) (*p* for non-linearity = 0.036). Abbreviations: SGA, small for gestational age; SI, serum iodine; OR, odds ratio.

**Table 1 nutrients-15-02868-t001:** Characteristics of pregnant women enrolled for serum iodine testing during early pregnancy, Huizhou Mother–Infant Cohort (n = 1101).

Variables	n	Mean ± SD, Median (*P*_25_–*P*_75_) or n (%)
Maternal age (y)	1101	28.1 ± 3.3
Pre-pregnancy BMI (kg/m^2^)	1101	20.8 ± 3.1
Gestational weight gain (kg)	719	13.5 ± 4.6
Excessive weight gain, n (%)	719	189 (26.3)
Insufficient weight gain, n (%)	719	219 (30.5)
Education level: university and above, n (%)	1101	272 (24.7)
Nulliparity, n (%)	1101	599 (54.4)
Smoking, n (%)	1077	28 (2.6)
Alcohol drinking, n (%)	1097	34 (3.1)
Hormone usage, n (%)	1099	245 (22.3)
Folate supplementation, n (%)	1100	1068 (97.1)
Iodine-fortified salt usage, n (%)	311	276 (88.7)
Habitual kelp and seaweed intake (g/d)	132	27.1 ± 40.0
Gestational diabetes, n (%)	968	164 (16.9)
Gestational hypertension, n (%)	809	16 (2.0)
PROM, n (%)	809	95 (11.7)
Full-term delivery (≥37 GWs), n (%)	808	771 (95.4%)
Abnormal amniotic fluid (<300 or >2000 mL), n (%)	589	76 (12.9)
Apgar scores at 1 min after delivery	809	9.83 ± 0.448
Medical and family history, n (%)	1099	
Family history of T2M among first-degree relatives		89 (8.1)
History of gestational diabetes		47 (4.6)
Family history of thyroid disorders		44 (4.0)
Any positive thyroid antibodies, n (%)		9 (0.8)
Serum iodine levels at T1 (μg/L)	1101	87.6 (77.4, 97.7)
Serum iodine levels at T2 (μg/L)	402	77.4 (71.1, 87.6)
Urinary iodine at T1 (μg/L)	389	190.6 (131.2, 260.0)
Urinary creatine at T1 (mg/L)	389	172.3 (114.4, 240.7)
UI/UCr at T1 (μg/g)	389	106.7 (76.7, 173.6)
Birth weight (kg)	809	3.16 ± 0.44
Birth length (cm)	809	49.9 ± 1.9
Ponderal index (kg/m^3^)	809	25.3 ± 2.0

Continuous variables are expressed as mean ± SD (normal distribution) or median (interquartile range, P_25_–P_75_) (skewed distribution). Categorical variables are expressed as n (%). Abbreviations: T2M, type 2 diabetes; UI/UCr, urinary iodine to urinary creatinine ratio; BMI, body mass index; T1, first trimester; T2, second trimester. Thyroid antibodies including antibodies of thyroperoxidase (TPOAb), TSH receptor (TRAb), and anti-thyroglobulin (TGAb) were measured only in women with abnormal thyroid function.

**Table 2 nutrients-15-02868-t002:** Odds ratios and 95% confidence intervals for the risk of selected gestational complications and adverse birth weight by quartiles of serum iodine concentrations during early pregnancy, Huizhou Mother–Infant Cohort (n = 1101).

	Quartiles of Serum Iodine Concentrations during Early Pregnancy (μg/L, Min~Max)				
	Q1 (45.68~77.41)	Q2 (78.68~87.56)	Q3 (88.83~97.71)	Q4 (98.98~229.69)	*P* _trend_
	n = 288	n = 281	n = 271	n = 261	
Gestational diabetes mellitus (GDM)					
Cases (%)	41 (15.47)	46 (18.04)	40 (15.94)	42 (18.03)	
Crude OR	1.0	1.202 (0.758~1.907)	1.036 (0.644~1.665)	1.201 (0.750~1.925)	0.600
Adjusted OR	1.0	1.323 (0.803~2.178)	1.135 (0.681~1.892)	1.476 (0.890~2.448)	0.215
Gestational hypertension (GH)					
Cases (%)	7 (3.3)	5 (2.4)	3(1.5)	1 (0.5)	
Crude OR	1.0	0.718 (0.224, 2.299)	0.431(0.110, 1.689)	0.156 (0.019–1.279)	0.062
Adjusted OR	1.0	0.803 (0.234–2.748)	0.471(0.112–1.987)	0.206 (0.024–1.754)	0.090
PROM					
Cases (%)	19 (9.0)	27 (13.0)	22 (10.7)	27 (14.4)	
Crude OR	1.0	1.586 (0.844, 2.981)	1.263 (0.656, 2.434)	1.724 (0.911, 3.262)	0.180
Adjusted OR	1.0	1.771 (0.920, 3.408)	1.397 (0.711, 2.747)	1.960 (1.010, 3.804)	0.103
Small for gestation age (SGA)					
Cases (%)	25 (11.9)	28 (13.5)	15 (7.3)	32 (17.1)	
Crude OR	0.719 (0.396, 1.305)	0.813 (0.458, 1.445)	0.385 (0.192, 0.771)	1.0	0.684
Adjusted OR	0.875 (0.458, 1.604)	0.855 (0.468, 1.563)	0.405 (0.198, 0.829)	1.0	0.887
Large for gestation age (LGA)					
Cases (%)	6 (2.9)	5 (2.4)	3 (1.5)	6 (3.2)	
Crude OR	1.105 (0.331, 3.690)	0.867 (0.247, 3.048)	0.546 (0.129, 2.322)	1.0	0.833
Adjusted OR	0.646 (0.179, 2.334)	0.607 (0.163, 2.262)	0.385 (0.085, 1.757)	1.0	0.958

Odds ratios and 95% CIs were estimated by multivariable logistic regression with covariates being included in the models by the enter method. For the outcome for gestational diabetes mellitus, the adjusted covariates included maternal age (years), education (primary school and below, junior and senior middle school/vocational high school, college, or university and above), pre-pregnancy BMI (kg/m^2^), nulliparity (yes or no), history of gestation diabetes mellitus (yes or no), family history of type 2 diabetes (yes or no), smoking (yes or no), and alcohol drinking (yes or no). For gestational hypertension, the adjusted covariates included maternal age (years), pre-pregnant BMI (kg/m^2^), nulliparity (yes or no), education (4 categories), smoking (yes or no), and alcohol drinking (yes or no). For PROM, women with infant birth weight > 4.0 kg or amino fluid volume > 2000 were excluded, 788 were included for analysis, and the adjusted variables included maternal age (years), pre-pregnant BMI (kg/m^2^), nulliparity (yes or no), education, smoking (yes or no), alcohol drinking (yes or no), delivery weeks, and birth weight. For SGA and LGA, the adjusted covariates included maternal age (years), pre-pregnancy BMI (kg/m^2^), nulliparity (yes or no), gestational weight gain (kg), delivery weeks, smoking (yes or no) and alcohol drinking (yes or no), and neonatal gender. Abbreviations: OR, odds ratio; SI, serum iodine; SGA, small for gestational age.

**Table 3 nutrients-15-02868-t003:** Associations of serum iodine concentrations during early pregnancy with selected obstetric and birth outcomes by multivariable linear regression analysis, Huizhou Mother–Infant Cohort.

	n	UnstandardizedCoefficients *B* (95% CI)	Standardized Coefficients *β*	*p*
Maternal log_10_ SIC at the first trimester (T1)				
Sum of Z-scores by OGTT	957	0.904 (−0.892, 2.700)	0.030	0.323
Apgar score at 1 min	806	0.448 (0.053, 0.844)	0.077	0.026
Gestational weight gain (kg)	715	−4.934 (−9.234, −0.634)	−0.082	0.025
Birth weight (kg)	715	−0.047 (−0.368, 0.274)	−0.008	0.775
Birth length (cm)	715	0.478 (−0.946, 1.903)	0.019	0.726
Ponderal index (kg/m^3^)	715	−1.002 (−2.819, 0.814)	−0.038	0.279
Maternal log_10_ SIC at the second trimester (T2)				
Sum of Z-scores by OGTT	388	−0.091 (−2.893, 2.711)	−0.003	0.949
Apgar score at 1 min	354	0.667 (0.010, 1.324)	0.105	0.047
Gestational weight gain (kg)	311	−13.006 (−20.084, −5.928)	−0.198	<0.001
Birth weight (kg)	311	−0.269 (−0.804, 0.267)	−0.043	0.325
Birth length (cm)	311	−0.333 (−2.715, 2.049)	−0.012	0.783
Ponderal index (kg/m^3^)	311	−1.183 (−4.219, 1.853)	−0.042	0.444
Maternal SIC change % from T1 to T2				
Sum of Z-scores by OGTT	388	−0.007 (−0.021, 0.007)	−0.048	0.319
Apgar score at 1 min	354	0.000 (−0.003, 0.003)	−0.004	0.937
Gestational weight gain (kg)	311	−0.043 (−0.080, −0.007)	−0.131	0.019
Birth weight (kg)	311	−0.003 (−0.005, 0.000)	−0.085	0.054
Birth length (cm)	311	−0.013 (−0.025, −0.001)	−0.092	0.038
Ponderal index (kg/m^3^)	311	−0.002 (−0.018, 0.013)	−0.017	0.753

Multivariable linear regression models were applied for data analysis with covariates being included by the enter method. The adjusted covariates for gestational weight gain included maternal age (years), education (five categories from below primary school to above university), nulliparity (yes or no), pre-pregnancy BMI (kg/m^2^), family history of diabetes (yes or no), delivery weeks (GWs), and alcohol drinking (yes or no). The adjusted covariates for birth weight, length, and ponderal index included maternal age (years), education (5 categories), prepregnant body mass index (kg/m^2^), delivery weeks (GWs), nulliparity (yes or no), neonatal gender (male or female), family history of diabetes (yes or no). The sum of Z-scores by OGTT was the sum of individual Z-scores of fasting, 1-, and 2-h plasma glucose levels by oral glucose tolerance testing (OGTT). The adjusted covariates for the Apgar score at 1 min included maternal age (years), education (5 categories), prepregnant body mass index (kg/m^2^), nulliparity (yes or no), delivery weeks (GWs), and birth weight (kg).

## Data Availability

The original datasets are not publicly available due to ethical considerations related to publishing medical record data, but are available from the corresponding author on reasonable request and in strict confidence.

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
