# Peer review of "Associations of Maternal Serum Iodine Concentration with Obstetric Complications and Birth Outcomes—Longitudinal Analysis Based on the Huizhou Mother–Infant Cohort, South China"

_nutrients, 2023, doi:10.3390/nu15132868_

Round 1

Reviewer 1 Report

The authors investigated the relationship between maternal serum iodine concentration in early pregnancy and obstetric complications and birth outcomes at delivery. It is an interesting topic, however, some questions were raised during the review process:

1.)    In the Methods part, statistical analysis you didn’t define from which level you consider significance: did you use p < 0.05 as usual or did you use other level of significance? When yes, please explain why it was chosen.

2.)    Quite confusingly, reading the Results part, authors report a significant relationship at p = 0.090 for a significantly elevated risk in one case (line 208-209), and a non-significant relationship at p = 0.054 in another (line 221). For those who are less familiar with odds ratios and only use the value of p as a guide, the current wording can be very misleading. Please try to rephrase it to make the significance clearer.

3.)    You measured UIC; was there any relationships between UIC or UI/UCr on the one hand and the obstetric complications and birth outcomes at delivery on the other hand either in total or divided in quartiles?

4.)    You made a dietary questionnaire about iodine consumption; could you find a relationship on the one hand dietary intake of iodine and on the other hand SIC, GDM, GH, SGA, LGA, birth weight overall and intake divided into quartiles? A recent study found a relationship between inadequate iodine intake and higher proportion of LGA newborns as well as macrosomia (Ovadia YS et al, 2022, https://doi.org/10.1007/s00404-021-06261-x).

5.)    In the Methods section (line 102) you write Ponderal Index was calculated with cubic of length (cm3) but in Table 1 unit of Ponderal index is kg/m3. Please correct it.

6.)    In m2, m3 the number should be in superscript. Please correct all of them in the whole manuscript.

Quality of English language is fine, only a few sentences have to be checked and rewritten to make sense.

Author Response

We are grateful for all the valuable and insightful comments from all the Reviewers.

The authors investigated the relationship between maternal serum iodine concentration in early pregnancy and obstetric complications and birth outcomes at delivery. It is an interesting topic, however, some questions were raised during the review process:

  • In the Methods part, statistical analysis you didn’t define from which level you consider significance: did you use p < 0.05 as usual or did you use other level of significance? When yes, please explain why it was chosen.

R:  Thanks for the Reviewer’s positive comments on our manuscript. The statistical significance level was set at p < 0.05. We have added this information in the section of statistical analysis (line 149, the clean version) as: SPSS 25.0, Amos 25, Medcalc and R 4.1.3 were applied for statistical analyses with significance being defined at p<0.05.

  • Quite confusingly, reading the Results part, authors report a significant relationship at p = 0.090 for a significantly elevated risk in one case (line 208-209), and a non-significant relationship at p = 0.054 in another (line 221). For those who are less familiar with odds ratios and only use the value of p as a guide, the current wording can be very misleading. Please try to rephrase it to make the significance clearer.

R:  Thanks for the comments and sorry for the confusions we have made. Although there was a marginally decreased trend for the risk of GH with the increasing quartiles of SIC, the overall association of SIC during early pregnancy with GH risk were negative. As suggested, we deleted the sentence in line 208-09 (original line number) and revised the sentence in line 201-03 (clear version) as: ‘After controlling for potential confounders, multivariable logistic regression (Table 2) showed null association between maternal SIC during early pregnancy with the risk of GDM, GH and LGA.

      The sentence in original line 221 (line 215-217, clean version) has been revised as: ‘Although null association was observed between maternal SIC at either T1 or T2 with birth weight and length, the increase of SIC change% from T1 to T2 was associated with decreased birth weight =-0.085, P=0.054) and length (β=-0.092, P=0.038) with marginal significance.

  • You measured UIC; was there any relationships between UIC or UI/UCr on the one hand and the obstetric complications and birth outcomes at delivery on the other hand either in total or divided in quartiles?

R: Thanks for the comments. we have already conducted the analysis on the associations of UI/UCr with various pregnancy outcomes, and all indicated non-significant findings (all P>0.05, data not shown). We have already described in the statistical analysis section that (line 178, clean version), ‘Additional analyses on the associations of UI/UCr with various pregnancy outcomes were also performed.’.

And in results part, we reported that (line 223-25, clean version), ‘Additional analyses on the associations of UI/UCr with various pregnancy outcomes, all indicated non-significant findings (all P>0.05, data not shown).’.

  • You made a dietary questionnaire about iodine consumption; could you find a relationship on the one hand dietary intake of iodine and on the other hand SIC, GDM, GH, SGA, LGA, birth weight overall and intake divided into quartiles? A recent study found a relationship between inadequate iodine intake and higher proportion of LGA newborns as well as macrosomia (Ovadia YS et al, 2022, https://doi.org/10.1007/s00404-021-06261-x).

R: Thanks for the comments. Due to resource limitation, we did not conduct detailed dietary survey, only two items in the questionnaire asked the participants about their habitual dietary kelp and seaweed intakes as well as the usage of iodized salt. As we described in Table 1 and also mentioned the limitation in the discussion, that only small portion of our participants (n=132) reported their habitual kelp and seaweed intakes. Additional linear regression analysis also indicates a negative correlation between dietary kelp and seaweed intakes with GWG, glucose levels, birth weight and length. Our analysis did not observe significant relation of SIC with LGA, GDM and GH. As compared with SIC, the bioavailable iodine to thyroid gland, dietary iodine estimation had large variations and did not consider the influence of digestion, absorption and metabolism of iodine in the human body.  

  • In the Methods section (line 102) you write Ponderal Index was calculated with cubic of length (cm3) but in Table 1 unit of Ponderal index is kg/m3. Please correct it.

R:  Sorry for the ignorance and typing errors. It should be cubed height by m3 not by cm3. We have carefully checked whole manuscript for the unit of PI and emended them accordingly.

6.)    In m2, m3 the number should be in superscript. Please correct all of them in the whole manuscript.

R:  Thanks. For all the m2 and m3, the numbers have been revised as superscript accordingly.

Comments on the Quality of English Language

Quality of English language is fine, only a few sentences have to be checked and rewritten to make sense.

R: We appreciate the positive comments from the Reviewer, and have checked all the sentences carefully to make no confusion.

References

  1. Mégier, C.; Dumery, G.; Luton, D. Iodine and Thyroid Maternal and Fetal Metabolism during Pregnancy. Metabolites 2023, 13, doi:10.3390/metabo13050633.
  2. Farebrother, J.; Zimmermann, M.B.; Abdallah, F.; Assey, V.; Fingerhut, R.; Gichohi-Wainaina, W.N.; Hussein, I.; Makokha, A.; Sagno, K.; Untoro, J., et al. Effect of Excess Iodine Intake from Iodized Salt and/or Groundwater Iodine on Thyroid Function in Nonpregnant and Pregnant Women, Infants, and Children: A Multicenter Study in East Africa. Thyroid : official journal of the American Thyroid Association 2018, 28, 1198-1210, doi:10.1089/thy.2018.0234.
  3. Rodriguez-Diaz, E.; Pearce, E.N. Iodine status and supplementation before, during, and after pregnancy. Best practice & research. Clinical endocrinology & metabolism 2020, 34, 101430, doi:10.1016/j.beem.2020.101430.
  4. Forhead, A.J.; Fowden, A.L. Thyroid hormones in fetal growth and prepartum maturation. The Journal of endocrinology 2014, 221, R87-r103, doi:10.1530/joe-14-0025.
  5. Jin, X.; Jiang, P.; Liu, L.; Jia, Q.; Liu, P.; Meng, F.; Zhang, X.; Guan, Y.; Pang, Y.; Lu, Z., et al. The application of serum iodine in assessing individual iodine status. Clinical endocrinology 2017, 87, 807-814, doi:10.1111/cen.13421.

Reviewer 2 Report

This is an interesting, large, complex study providing some new information with potential clinical benefit. However, the outcomes reported are difficult to readily accept, based upon the assumptions provided by the authors, and will need to be confirmed by other independent studies with different research designs. The "U shaped curve: in Figure 2 is very interesting. A summary of your paragraph on the limitations of this study should be included in the Abstract.

I could not see the data correleting UIC and TFTs (especially serum TSH) with SIC. Further, the authors do not provide any comments on the correlation between UIC and TFTs with pregnancy and fetal outcomes. Are these data being saved for another publication or have they not been processed, or did I miss it?

Did the women habitually eating kelp/seaweed have higher SIC concentrations? Did you analyse their data as a separate group.

Could you please explain or correct the title of your manuscript. What does the word "rition" mean?

Line 73 should be "embedded" not "imbedded".

There are some typographical errors in the Table. The sentences on lines 324 and 325 need to be corrected.

Author Response

We are grateful for all the valuable and insightful comments from all the Reviewers. 

This is an interesting, large, complex study providing some new information with potential clinical benefit. However, the outcomes reported are difficult to readily accept, based upon the assumptions provided by the authors, and will need to be confirmed by other independent studies with different research designs. The "U shaped curve: in Figure 2 is very interesting. A summary of your paragraph on the limitations of this study should be included in the Abstract.

R: Thanks for the Reviewer’s positive comments and insightful suggestion on our work. Current study was conducted among generally healthy and euthyroid women with the main purpose for establishment of population and trimester-specific reference intervals of serum iodine concentrations. We agree with the Reviewer that, the conclusions should be confirmed in further research by different study design. As we have elaborated in both discussion and conclusions that:
1) Discussion (line 420-22, clean version): ‘Additional studies with inclusion of mothers of thyroid dysfunction and adverse birth outcomes are warranted to substantiate current findings and extend the study generalizability.’.
2) Conclusions (line 434-36, clean version): ‘Future longitudinal studies with inclusion of mothers of various thyroid dysfunction and obstetric complications and follow-up for infant and childhood neurodevelopment are warranted.’

Actually, we have increased the samples for SIC testing at different trimesters with inclusion of mothers of various thyroid conditions or adverse obstetric outcomes. And several 1:1 matched nested several case-control studies have been imbedded in the cohort with aims to testify/confirm the relationship on SIC across gestation with various pregnancy outcomes esp. the adverse birth outcomes, such as SGA and LGA etc.

Yes, the U-shaped association of SIC at T1 with SGA risk was interesting. As reminded, we include this finding in abstract accordingly as: Restricted cubic spline regression suggested U-shaped association between SIC with SGA risk, and SIC above 94μg/L at T1 initiated with increased risk of SGA. We are also planning to confirm this finding in the coming nested case-control study. As the journal of Nutrients had word limit for abstract for less than 200 words. It might be not eligible to include the limitations in the abstract.

I could not see the data correlating UIC and TFTs (especially serum TSH) with SIC. Further, the authors do not provide any comments on the correlation between UIC and TFTs with pregnancy and fetal outcomes. Are these data being saved for another publication or have they not been processed, or did I miss it?

R:  Thanks for the comments. I am sure about your abbreviation for TFT. Does that mean thyroid hormones as you mentioned TSH in the following bracket? The relationship of UIC, thyroid markers and SIC have been reported in another manuscript which are currently under review process by another journal. And we found stronger associations of SIC with thyroid hormones than those of UIC even UCr corrected UIC.

Did the women habitually eating kelp/seaweed have higher SIC concentrations? Did you analyze their data as a separate group.

R: Thanks. we only had a small portion of participants responded to this question. As suggested, spearmen correlation analysis was performed while the findings suggested null correlation between them.

Comments on the Quality of English Language

Could you please explain or correct the title of your manuscript. What does the word "rition" mean?

R: Thanks a lot. We are very sorry the typing error in the title of the manuscript. We deleted the wrong characters and the title has been revised as: ‘Associations of Maternal Serum Iodine Concentrations with Obstetric Complications and Birth Outcomes- Longitudinal analysis on the basis of Huizhou Mother-infant Cohort, South China.’

Line 73 should be "embedded" not "imbedded".

R: Have revised. Many thanks.

There are some typographical errors in the Table. The sentences on lines 324 and 325 need to be corrected.

R: Thanks for pointing out the errors for improvement of our work. We have carefully checked all the main text and tables, and revisions have been made accordingly (please kindly find the revised version with track of changes for the information). The sentence in line 324-25 was: Our study emphasized the crucial importance of SIC surveillance during gestation esp. in mid-trimester among a population known to be susceptible for both iodine deficiency and excess.

Reviewer 3 Report

In their paper Liu et al have addressed the question, whether serum iodine concentrations (SIC) impact on gestational parameters and infant outcome. This is important, as iodine is an essential, and in many populations critical, nutrient. Nevertheless, there are several questions and limitations to be addressed.

Major:

1. Introduction, l. 41ff: A single review is not adequate here. Please go for original articles!!

2. l. 48: original literature please!

3. l. 64, ref 10: This paper points to the methods, no to the medical value of this method.

4. l. 69: 'to assess iodine nutrition': SIC cannot be a marker of nutrition, but only be a proxy of iodine status. A marker of iodine nutrition is calculation from precise food protocols!

5. l. 78: '...aged 18-45 years' does not match the age in the flow sheet (Fig, 1) (20-34y)! Moreover, define the collection time period and total numbers of deliveries in that time and region. Is the study population representative for the whole community?

6. l 95f: Provide the questionnaire in English in the online supplement, please!

7. l. 190/Table 1: According to the values of SIC, the value range in the 4. quartile is much higher than in the other groups, possibly showing a subgroup of extremely high values. Hence, this 4. quartile may comprise a simple 'high value' and a 'toxicity' sub-group. The reviewer suggests to particularly address these very high level individuals, e. g. by forming interquintile rather than interquartile groups to avoid confounding.

8. l. 219: A non-significant relationship isn't a relationship!

9. l. 220, SIC change %: Unclear wording, whether this means an increase or decrease or any change direction.

10. l. 320: Individual iodine status was not shown in this study. The study shows useful associations between SIC and gynocological/neonatological risks and outcomes, nothing more!

11. l.323: ' ...prescription of iodine supplements during pregnancy...' Definitely not! This paper was submitted to Nutrients, a nutrition journal, wasn't it? Adequate intake of essential nutrients isn't primarily a matter of blister packs! It is a matter of nutrition, and of avoiding inadequate exposures.

12. l. 427-431: This argumentation is not correct. Underpowering is a matter of sample size, excluding still births and aborts is a confounder of  inclusion parameters.

Minor:

1. L. 24, fasting venous >blood< samples

2. L. 25ff, introduce abbreviations at first mentioning.

3. l. 57 change ... 'urinary creatinine' to 'urinary creatinine was performed'..

4. l. 58: replace 'pregnant' by 'pregnancy' and 'recent' by 'actual'.

5. l. 59: formatting of references.

6. l. 108: For defining centrifugation, indicate g force rather than rpm and provide temperature, please.

7. l. 118: Adjust spaces!

8. l. 253ff: p value missing. What do the authors want to say in the sentence?

9. l. 337 + 345: grammar of sentences.

10. l. 354-357: Invert sequence of the 2 sentences.

11. l. 379: 'the large decrease'. Do the authors mean the absolute or fractional decrease?

12, l. 384f : reference missing.

The English should be checked. In general, the ms is well written, but there are many grammar errors.

Author Response

We are grateful for all the valuable and insightful comments from all the Reviewers.

In their paper Liu et al have addressed the question, whether serum iodine concentrations (SIC) impact on gestational parameters and infant outcome. This is important, as iodine is an essential, and in many populations critical, nutrient.

R: Thanks a lot for the positive comments from the Reviewer.

Nevertheless, there are several questions and limitations to be addressed.

Major:

  1. Introduction, l. 41ff: A single review is not adequate here. Please go for original articles!!

R: Thanks for the suggestion. We have added several references mostly original papers in the first paragraph of introduction.

Iodine is one essential ingredient of thyroid hormones. Iodine requirements during gestation increase sharply due to elevated hormone production, fetal needs, and renal excretion[1]. Iodine above adequacy or excess becomes prevalent in recent years and attracts a lot of concerns because of the extensive iodine prescription during gestation and universal salt iodization even in iodine-replete regions[2]. Pregnant women and neonates are susceptible to both iodine deficiency and excess as both of which were associated with poor pregnancy outcomes[3]. Early pregnancy is a particularly vulnerable period to iodine nutrition since maternal iodine is the only source of fetal thyroid hormone synthesis[4]. However, the potential influence of excessive iodine exposure has not been well studied especially during gestation.

  1. 48: original literature please!

R:  In line 48, the reference 2 has been modified as [4]: Forhead AJ, Fowden AL. Thyroid hormones in fetal growth and prepartum maturation. J Endocrinol. 2014 Jun;221(3):R87-R103. doi: 10.1530/JOE-14-0025. Epub 2014 Mar 19. PMID: 24648121.

  1. 64, ref 10: This paper points to the methods, no to the medical value of this method.

R: Thanks for the advice. The reference has been changed as [5]: Jin, X.; Jiang, P.; Liu, L.; Jia, Q.; Liu, P.; Meng, F.; Zhang, X.; Guan, Y.; Pang, Y.; Lu, Z., et al. The application of serum iodine in assessing individual iodine status. Clinical endocrinology 2017, 87, 807-814, doi:10.1111/cen.13421.

  1. 69: 'to assess iodine nutrition': SIC cannot be a marker of nutrition, but only be a proxy of iodine status. A marker of iodine nutrition is calculation from precise food protocols!

R: Thanks. We agree with the Reviewer that dietary iodine intake is one of the most essential components for assessing iodine nutrition. However, dietary survey is limited by information bias or recall bias. Biochemical testing esp. SIC has been approved to be a valid biomarker for assessing individual iodine status. In addition, health examination and clinical symptoms related with iodine inadequacy and excess are also important for evaluating iodine nutrition.  The sentence in line 67-69 (clean version) has been revised as: ‘We used trimester-specific SIC as clinical biomarker to assess individual iodine status for pregnant women and explore their relationship with common obstetric complications and birth outcomes.’

  1. 78: '...aged 18-45 years' does not match the age in the flow sheet (Fig, 1) (20-34y)! Moreover, define the collection time period and total numbers of deliveries in that time and region. Is the study population representative for the whole community?

R: Thanks for the comments and sorry for the confusions we have brought. The discrepancy in maternal age range was due to the different inclusion criteria for different study purposes, one age range (18-45 years) was for participants’ recruitment to establish Huizhou Mother-infant cohort, and the other (20-34) was for establishment reference intervals of SIC among generally euthyroid pregnant women. We thus revise the sentence in line 78-80 as: ‘Singleton pregnant women aged 20-34 years of regular residence in Huizhou city were selected from Huizhou Mother-infant Cohort during their first antenatal visit from September 2020 to June 2021 with original purpose for establishment of trimester-specific reference intervals for SIC.’

  1. l 95f: Provide the questionnaire in English in the online supplement, please!

R:  Thanks for the comments. However, the questionnaire was only for data collection of basic information which was only applicable in Chinese population. As we have described in methodology (line 92-96, clean version): ‘Individual information including socio-demographics, health and obstetric history, family history of common chronic diseases, medication treatment and lifestyle factors (i.e., smoking, alcohol drinking, physical activities, usage of iodine fortified salt, and dietary consumption of iodine-rich foods) were collected via face-to-face interview using a pretested questionnaire.’. We appreciate the reviewer’s suggestion. The questionnaire will be translated into English and submitted as supplemental material if reqested by the Editorial office.

  1. 190/Table 1: According to the values of SIC, the value range in the 4. quartile is much higher than in the other groups, possibly showing a subgroup of extremely high values. Hence, this 4. quartile may comprise a simple 'high value' and a 'toxicity' sub-group. The reviewer suggests to particularly address these very high level individuals, e. g. by forming interquintile rather than interquartile groups to avoid confounding.
  2. Thanks for the comments. In line 190-92 (clean version), we have described the baseline information of participants in Table 1. The median (IQR, P25 to P75) SIC at 10-12 GWs (T1) and 24-28 GWs (T2) were 87.6 (77.4, 97.7) and 77.4 (71.6, 87.6) μg/L, respectively. The size of interquartile ranges of SIC at T1 in the fourth quartile was similar with other quartile groups: 65.99-74.87 μg/L for Q1; 81.22-86.29 μg/L for Q2; 90.10-95.18 μg/L for Q3 and 101.52-111.67 μg/L for Q4. We have checked all the SIC data, in the fourth quartiles of SIC, there were 4 values at T1 and 1 value at T2 were above 150 ug/L (> mean+3 SD). However, they would not affect the overall conclusions as we applied quartiles of SIC in the logistic regression models and logarithmic transformation for SIC in the linear regression models. In addition, multivariable adjusted regression of Restricted Cubic Spline (RCS) was explored to visualize the relationship of maternal SIC at T1 with the risk of SGA. The risk of SGA was decreased with the increase of maternal SIC (T1) until reaching 94 μg/L, and then gradually increased thereafter.

  1. l. 219: A non-significant relationship isn't a relationship!

R: Thanks. The sentence in line 215 (clean version) has been revised as: ‘Although null association was observed between maternal SIC at either T1 or T2 with birth weight and length…’

  1. 220, SIC change %: Unclear wording, whether this means an increase or decrease or any change direction.

R: Thanks for the comments. The sentence in line 215-218 (clean version) has been therefore revised as: ‘Although null association was observed between maternal SIC at either T1 or T2 with birth weight and length, the increase of SIC change% from T1 to T2 was associated with decreased birth weight =-0.085, P=0.054) and length (β=-0.092, P=0.038) with marginal significance.

  1. 320: Individual iodine status was not shown in this study. The study shows useful associations between SIC and gynocological/neonatological risks and outcomes, nothing more!

R: We can not catch up the reviewer’s comments. The original data of the study will be available from the corresponding author if necessary. As we have claimed in the Data Availability Statement that (line 469-71): ‘The original datasets are not publicly available due to ethical limitations in publishing medical record data, but available from the corresponding author on reasonable request under strict confidential process.

The original study purpose was to establish the reference intervals for SIC during gestation. Majority of our participants were euthyroid. The study implication lies in that, even in euthyroid mothers, we still have observed meaningful associations between SIC and pregnant outcomes.

  1. 323: ' ...prescription of iodine supplements during pregnancy...' Definitely not! This paper was submitted to Nutrients, a nutrition journal, wasn't it? Adequate intake of essential nutrients isn't primarily a matter of blister packs! It is a matter of nutrition, and of avoiding inadequate exposures.

R: Yes. We agree with the Reviewer that supplementation of nutrients is only necessary under inadequate exposure. As we explained in introduction, pregnant women are vulnerable to both iodine deficiency and excess. Our study observed an adverse influence on birth outcomes with high iodine status. The sentence in line 313-14 (clean version) has been revised as: ‘The findings are useful for healthcare providers for reasonable prescription of iodine supplements during pregnancy to avoid excessive iodine exposure.’.

  1. 427-431: This argumentation is not correct. Underpowering is a matter of sample size, excluding still births and aborts is a confounder of inclusion parameters.

R: Thanks for the comments. With the original purpose for establishment of reference ranges for SIC during gestation, we chose pregnant women of generally healthy and euthyroid for SIC testing. Women of still births or pregnant loss were excluded for analysis.

We therefore revised the sentence in limitation of discuss section (line 419-20, clean version) as: ‘First, most of our participants were healthy and euthyroid. Additional studies with inclusion of mothers of thyroid dysfunction and adverse birth outcomes are warranted to substantiate current findings and extend the study generalizability.

Minor:

We appreciate the Reviewer for the careful comments on our article. All the minor comments have been considered and revised accordingly.

  1. 24, fasting venous >blood< samples

R: Thanks. have revised.

  1. 25ff, introduce abbreviations at first mentioning.
  2. Have revised. Thanks.
  3. l. 57 change ... 'urinary creatinine' to 'urinary creatinine was performed'..

R: Have revised. Thanks.

  1. 58: replace 'pregnant' by 'pregnancy' and 'recent' by 'actual'.

R: Thanks, have revised.

  1. l. 59: formatting of references.?

R: Thanks. We have checked and reformatted all the references.

  1. l. 108: For defining centrifugation, indicate g force rather than rpm and provide temperature, please.

R: Thanks. Assumed a radius of 13.5 cm for common sigma instruments for centrifugation, the, the force of centrifugation was 1349 g based on the formula of G=1.11×10^(-5)×R×(rpm)^2. We have revised the description accordingly.   

  1. l. 118: Adjust spaces!

R: Have deleted the unnecessary spaces between words.

  1. l. 253ff: p value missing. What do the authors want to say in the sentence?
  2. Sorry we are unclear about the reviewer’s comments, for where the p values were missing? For the results of logistic regression, we reported OR and 95%CI, to save the table space, we did not indicate p values simultaneously. Like most of publications, it is not obligatory to report P values if 95%CI has been provided.

  1. l. 337 + 345: grammar of sentences.

R: Thanks. The sentences in line 328-330(clean version) have been revised as: The non-significant findings might be due to the intrinsically high variability of UIC than that those of SIC which caused inadequate statistical power and compromised the comparisons between groups for UIC analysis.

  1. l. 354-357: Invert sequence of the 2 sentences.

R: The sentence has been revised as: High iodine level was reported to be associated with reduced oxidative stress and lowered risk of GH[22], as iodine might contribute to redox balance as anti-oxidant during pregnancy.

  1. 379: 'the large decrease'. Do the authors mean the absolute or fractional decrease?

R: Thanks. It referred to the relative change, that is, the change percentage of SIC from T1 to T2. The sentence has been revised as: Our findings indicated that higher maternal SIC at both early and mid-trimester and even the relatively less decrease of SIC during gestation could restrict maternal weight gain with a highest impact being observed for mid-trimester SIC.

12, l. 384f : reference missing.

R: The Ref.31 (original 29) was indicated in line 377 (clean version).

Comments on the Quality of English Language

The English should be checked. In general, the ms is well written, but there are many grammar errors.

R:  Thanks a lot for the reviewer’s positive comment. The manuscript has been carefully checked for the grammar errors, and all the references were checked and some updated. All the changes in the revised version have been marked with the function of track of change.

References

  1. Mégier, C.; Dumery, G.; Luton, D. Iodine and Thyroid Maternal and Fetal Metabolism during Pregnancy. Metabolites 2023, 13, doi:10.3390/metabo13050633.
  2. Farebrother, J.; Zimmermann, M.B.; Abdallah, F.; Assey, V.; Fingerhut, R.; Gichohi-Wainaina, W.N.; Hussein, I.; Makokha, A.; Sagno, K.; Untoro, J., et al. Effect of Excess Iodine Intake from Iodized Salt and/or Groundwater Iodine on Thyroid Function in Nonpregnant and Pregnant Women, Infants, and Children: A Multicenter Study in East Africa. Thyroid : official journal of the American Thyroid Association 2018, 28, 1198-1210, doi:10.1089/thy.2018.0234.
  3. Rodriguez-Diaz, E.; Pearce, E.N. Iodine status and supplementation before, during, and after pregnancy. Best practice & research. Clinical endocrinology & metabolism 2020, 34, 101430, doi:10.1016/j.beem.2020.101430.
  4. Forhead, A.J.; Fowden, A.L. Thyroid hormones in fetal growth and prepartum maturation. The Journal of endocrinology 2014, 221, R87-r103, doi:10.1530/joe-14-0025.
  5. Jin, X.; Jiang, P.; Liu, L.; Jia, Q.; Liu, P.; Meng, F.; Zhang, X.; Guan, Y.; Pang, Y.; Lu, Z., et al. The application of serum iodine in assessing individual iodine status. Clinical endocrinology 2017, 87, 807-814, doi:10.1111/cen.13421.

Round 2

Reviewer 1 Report

The authors revised the manuscript according to the reviewers’ comments and it has improved a lot. I accept the answers to all my questions. Some minor suggestions, corrections:

1.)    Abstract, line 32: delete one full stop at the end of the sentence.

2.)    line 395 please change ul to μl.

3.)    Line 420 full stop is missing after euthyroid.

4.)    Line 440 Author contributions: You write that ‘YW made similar contribution to the manuscript.’ Similar to whom, which author? Please correct it.

The English is fine, only some spellcheck is needed. Some typos or spelling mistakes that should be corrected:

1.)    Line 190: it sound better if you write half of them (mothers) were primipara ( or primiparous; instead of prim-parity).

2.)    Line 273: change letter ‘e’ to ‘u’ in mellitus.

3.)    Line 336-8: rewrite the sentence to: ‘This might be due to the fact that the… appeared to be more likely to develop GDM.

4.)    Line 338 instead of ‘a positive association’ you should write ‘positive associations’. Please correct it.

5.)    Line 382 ‘A study of 462 Belgian mothers….’ Please correct it.

6.)    Line 388 ‘in’ is missing from ‘in utero environment’

Author Response

The authors revised the manuscript according to the reviewers’ comments and it has improved a lot. I accept the answers to all my questions.

  1. Thanks a lot for the Reviewer’s positive comments and kind acceptance of our responses.

Some minor suggestions, corrections:

  • Abstract, line 32: delete one full stop at the end of the sentence.

R: Have deleted. Thanks.

  • line 395 please change ul to μ

 R: Have revised, thanks.

3.)    Line 420 full stop is missing after euthyroid.

R:  Thanks.

4.)    Line 440 Author contributions: You write that ‘YW made similar contribution to the manuscript.’ Similar to whom, which author? Please correct it.

R: have revised as: YW made similar contribution to the manuscript with the first author.

Comments on the Quality of English Language

The English is fine, only some spellcheck is needed. Some typos or spelling mistakes that should be corrected:

  • Line 190: it sound better if you write half of them (mothers) were primipara ( or primiparous; instead of prim-parity).

R: Thanks.

  • Line 273: change letter ‘e’ to ‘u’ in mellitus.

R: Thanks. Change has been made as suggested.

  • Line 336-8: rewrite the sentence to: ‘This might be due to the fact thatthe… appeared to be more likely to develop GDM.

R: Thanks a lot. The sentence has been revised.

  • Line 338 instead of ‘apositive association’ you should write ‘positive associations’. Please correct it.

R: Thanks.

  • Line 382 ‘A study of462 Belgian mothers….’ Please correct it.

R: Thanks. Have revised

  • Line 388 ‘in’ is missing from ‘in utero environment’

R: Thanks. Have revised.

Reviewer 3 Report

It's fine now.

Author Response

Thanks a lot for the Reviewer's acceptance of our responses. 

Round 3

Reviewer 3 Report

It's fine now.

Author Response

Thanks.